# Pathogen Detection in *Ornithodoros sonrai* Ticks and Invasive House Mice *Mus musculus domesticus* in Senegal

**DOI:** 10.3390/microorganisms10122367

**Published:** 2022-11-30

**Authors:** Basma Ouarti, Moussa Sall, El Hadji Ibrahima Ndiaye, Georges Diatta, Adama Zan Diarra, Jean Michel Berenger, Cheikh Sokhna, Laurent Granjon, Jean Le Fur, Philippe Parola

**Affiliations:** 1Aix-Marseille University, IRD, SSA, AP-HM, VITROME, 13005 Marseille, France; 2IHU-Méditerranée Infection, 13005 Marseille, France; 3UFR des Sciences Appliquées et de Technologie, Section Informatique, University Gaston Berger, Saint-Louis 32000, Senegal; 4Institut de Recherche Pour le Développement (IRD), VITROME IRD 257, Campus International IRD-UCAD, Hann, Dakar 1386, Senegal; 5Centre de Biologie Pour la Gestion des Populations, Institut de Recherche Pour Développement, Centre de Coopération Internationale en Recherche Agronomique Pour le Développement, Institut National de Recherche Pour l’Agriculture, l’Alimentation et l’Environnement, Institut Agro Montpellier, University Montpellier, Lez CEDEX, 34988 Montferrier, France

**Keywords:** *Borrelia crocidurae*, *Ornithodoros sonrai*, *Mus musculus domesticus*, Anaplasmataceae, Senegal

## Abstract

*Ornithodoros sonrai* (*O. sonrai*) ticks are the only known vectors of *Borrelia crocidurae*, an agent of tick-borne relapsing fever (TBRF) borreliosis. Rodents serve as principal natural reservoirs for *Borrelia*. Our research objective was to detect TBRF *Borrelia* and other zoonotic bacterial infections in ticks and in house mice *Mus musculus domesticus*, an invasive species currently expanding in rural northern Senegal. Real-time and conventional PCR were utilized for detecting *Borrelia* and other bacterial taxa. The analyses were performed on 253 specimens of *O*. *sonrai* and 150 samples of brain and spleen tissue from rodents. *Borrelia crocidurae* was found in one *O. sonrai* tick and 18 *Mus musculus domesticus* samples, with prevalences of 0.39 percent and 12 percent, respectively, as well as *Ehrlichia* sp. in one *Mus musculus domesticus*. Further, we were able to detect the presence of a potentially infectious novel species belonging to the Anaplasmataceae family for the first time in *O. sonrai* ticks. More attention should be paid to the house mouse and *O. sonrai* ticks, as they can be potential hosts for novel species of pathogenic bacteria in humans.

## 1. Introduction

Ticks are strict blood-sucking arthropods divided into two families, *Ixodidae* (about 700 hard tick species) and *Argasidae* (around 200 soft tick species). The latter comprises the sub-families Argasinae and Ornithodorinae, which include the genus *Ornithodoros* [1,2]. Ticks can have a negative impact on an animal’s health. They represent the second largest group for the transmission of infectious pathogens to humans after mosquitoes [3]. Tick-borne relapsing fever (TBRF) borreliosis is caused by various species of spirochetes of the genus *Borrelia*, of which *Borrelia crocidurae* (Léger, 1917)*, Borrelia duttonii* (Novy and Knapp, 1906) and *Borrelia hispanica* (de Buen, 1926) occur in Africa. These spirochetes are transmitted by soft ticks of the genus *Ornithodoros* [4,5,6]. As the name implies, TBRF is characterized by recurrent fever and associated symptoms, and in extreme cases the infection can spread to the neurological and cardiovascular systems [6]. In West Africa, TBRF is endemic in the Sahara, Sahel and Sudano-Sahelian regions [5]. In West and North Africa, small mammals are considered to be the principal reservoirs of *Borrelia*, particularly of *B. crocidurae* [7], which has been detected in several rodent species [5,7,8]. *Borrelia crocidurae* is transmitted via a bite by the tick *Ornithodoros sonrai* (Sautet and Witkowski, 1943 [5,7]) that lives inside small rodent burrows [9].

In Senegal, TBRF is a major cause of diseases, although it is frequently misdiagnosed (and often confused with malaria) by clinicians due to a lack of diagnostic tools [5]. Several studies conducted in the country have reported the presence of *B. crocidurae* in ticks and/or small mammals [5,7,8,9,10], with a prevalence ranging from 4.54% to 72.72% for *Borrelia* spp. and *B. crocidurae*, respectively [7]. A large-scale screening in North and West Africa detected the presence of *B. crocidurae* in *O. sonrai* ticks; *B. hispanica* in *Ornithodoros marocanus* (Velu, 1919), *Ornithodoros occidentalis* (Trape, Diatta, Durand and Renaud, 2013) and *Ornithodoros kairouanensis* Trape, (Diatta, Bouattour, Durand and Renaud, 2013); and *Borrelia merionesi* (Blanc and Maurice, 1948) in *Ornithodoros merionesi* (Trape, Diatta, Belghyti, Sarih, Durand and Renaud, 2013) and *Ornithodoros costalis* (Diatta, Bouattour, Durand, Renaud and Trape, 2013) [5]. Other recent studies in rural areas of Senegal have also shown the presence of (i) *Coxiella burnetii* (Derrick, 1939), the causative agent of Q fever, in *O. sonrai* [11]; (ii) several bacteria in small rodent spleen tissues, namely *Bartonella* spp. and Anaplasmataceae with potential new species Candidatus “*Ehrlichia senegalensis*” (Dahmana, Granjon, Diagne, Davoust, Fenollar, Mediannikov, 2020) and *Borrelia* spp., with a prevalence of 9.35%, 18.12% and 15.2%, respectively [9].

The purpose of this study was to gain a better understanding of the risk associated with the transmission of TBRF and other zoonotic bacteria carried by invasive small rodents in the rural area of Dodel, northern Senegal, and more particularly the domestic mouse *Mus musculus domesticus* (Schwarz and Scharz, 1943) [12]. We also sought to identify an expanded repertoire of infectious agents in order to increase the awareness of pathogens transmitted by *O. sonrai* soft ticks, hosted by small rodents, which may have a negative impact on human and animal health.

## 2. Materials and Methods

### 2.1. Study Area

The study was conducted in the rural commune of Dodel (16°29′16.7″ N/14°25′19.0″ W), located 531 km northeast of Dakar, in the Saint-Louis region (Figure 1). In this area, the vegetation and climate are typically Sahelian, with a short wet season from July to mid-October. The survey included 60 housing plots visited for rodents and 34 for *Ornithodoros* ticks in 2018, as well as 77 and 10 others, respectively, in 2021 in Dodel’s central area. These houses accommodate 489 people, the majority of them belonging to the Peulh “Pulaar” ethnic group. The population is more involved in trade than in agriculture and livestock.

### 2.2. Sampling

Rodents were captured alive using associated sets of locally-made wire-mesh and Sherman (H.B. Sherman Traps, Inc., Tallahassee, FL, USA) folding box traps placed within dwellings (bedrooms, hallways, kitchens, etc.) and shops. Trapping campaigns within private dwellings were conducted with prior explicit oral agreements from the relevant local authority (head of village and mayor) and individual householders after providing a comprehensive presentation of the project and its objectives. The traps were baited daily with groundnut paste and left for one to three days [7,9]. They were checked every morning and afternoon (while re-baiting) for night and day captures, respectively. All specimens caught were identified according to morphological criteria as previously reported [13]. Rodents were euthanized by performing cervical dislocation and handled in accordance with the relevant requirements of Senegalese legislation and the live animal capture guidelines of the American Society of Mammalogists (Animal Care and Use Committee 2011). Body measurements were taken before autopsy and reproductive status was noted. Brain and spleen tissue as well as digestive tracts were preserved in 95% ethanol for further analyses. All protocols used here were conducted following official regulations (Centre de Biologie pour la Gestion des Populations (CBGP)): Agrément pour l’utilisation d’animaux à des fins scientifiques D-34-169-003) of the relevant institutional committee (Regional Head of the Veterinary Service, Hérault, France). All transfer and conservation procedures were performed in accordance with current Senegalese and French legislation.

Overall, 150 rodents (59 in 2018 and 91 in 2021) were captured over 966 trap-nights, leading to an overall trapping success of 15.22%. They belonged to two species of the family Muridae, i.e., *Mus musculus domesticus* (147 individuals) and *Arvicanthis niloticus* (É. Geoffroy, 1803) (3 individuals).

*Ornithodoros* ticks were collected from inside rodent burrows and crevices identified in domestic and peri-domestic environments using a modified portable petrol-powered aspirator [5]. They were kept in cryotubes containing 95% ethanol for the identification of their morphological features and further molecular analyses. A total of 104 rodent burrows were examined in dwellings and peri-domestic areas. The rodent burrow infestation rate was calculated as the ratio of the number of burrows harboring ticks to the total number of burrows sampled.

### 2.3. Ornithodoros Tick Identification and DNA Extraction

The morphological identification of *Ornithodoros* ticks was carried out based on a tick identification guide [1]. Observations were carried out using an optical microscope, (Zeiss Axio Zoom V16 (Zeiss, Marly-le-Roi, France)), and a scanning electron microscope (SEM) (Hitachi High-Technologies, Tokyo, Japan).

DNA extraction was performed on a small portion of spleen and brain tissue separately for each rodent, as well as on ticks, using an EZ1 DNA tissue extraction kit (Qiagen, Hilden, Germany), and following a previously described protocol [9,14,15]. All *Ornithodoros* ticks were subjected to standard PCR and sequencing to confirm morphological identification using the *16S* tick gene [14].

### 2.4. Screening for Bacteria

Bacteria screening was performed on rodent brain and spleen tissue as well as on *Ornithodoros* ticks using a real-time PCR (qPCR) CFX96 system (Bio-Rad, Marnes-la-Coquette, France). Bacterial detection was applied to each sample to detect the presence of bacteria such as *Anaplasma*, *Borrelia*, *Bartonella*, *C. burnetii* and *Rickettsia* using specific primers and probes [16]. Positive and negative controls were used in each molecular reaction as previously described [16]. Samples with a cycle threshold of (Ct) < 36 were considered positive [16,17]. The DNA of samples positive in qPCR was amplified using standard PCR, then sequenced to identify the bacterial species [16,18]. The genes used to target the presence of Anaplasmataceae and *Borrelia* spp. in rodents and/or *O. sonrai* ticks were Anaplasmataceae *23S* rRNA [16] and *Borrelia 16S* rRNA [19]. Only the samples from rodents captured in 2018 were tested for Anaplasmataceae. Samples positive for the Anaplasmataceae *23S* rRNA gene were tested with an additional PCR specific to the genus *Ehrlichia groEL* [20] and the Anaplasmataceae *16S* rRNA gene [21], and those positive for *Borrelia* spp. based on *16S_Bor* rRNA were subjected to qPCR for the *glp*Q gene, specific to *B. crocidurae*, as previously described [10]. Once the PCR products were amplified, electrophoretic migration was performed for each PCR reaction in a 1.5% agarose gel stained with SYBR Safe TM (Thermo Fisher Scientific, Illkirch, France) and visualized using the ultraviolet imager ChemiDoc MP (Bio-Rad, Marnes-la-Coquette, France). The PCR products were purified and sequenced using a Big Dye Terminator (Thermo Fisher Scientific, Illkirch, France) and genetic analyzer ABI PRISM 3130 (Applied Biosystems, Courtaboeuf, France) [16] with ChromasPro software, version 1.34 (Technelysium Pty, Ltd., Tewantin, QL, Australia). The sequences obtained were analyzed and assembled for comparison with sequences from the GenBank database using the BLAST tool, as described previously [9,16]. The phylogenetic analyses were established using MEGA software, version 7.0.21, with 100 bootstrap replications [22]. The accession numbers assigned by GenBank to all our sequences are listed as follows: for the Anaplasmataceae *23S* rRNA gene (*Anaplasma* sp. (MW790939) and *Ehrlichia* sp. (MW790940)); for the Anaplasmataceae *16S* rRNA gene (*Anaplasma* sp. (MW790941) and *Ehrlichia* sp. (MW790942)) and finally for the gene *Ehrlichia groEL* (*Ehrlichia* sp. (MW791746)).

## 3. Results

### 3.1. Sampling of Rodents and Ornithodoros Ticks

A significant difference in the relative abundance of invasive house mice was observed between the two capture sessions in 2018 and 2021, with trapping success values of 11.4% (59 individuals/518 trap-nights) and 20.3% (91/448), respectively (Chi-square test = 10.6175; *p* = 0.00112). Otherwise, an overall burrow infestation rate of 22.1% was recorded (23/104 rodent burrows were infested with *O. sonrai* ticks), leading to the collection of 253 specimens of ticks.

### 3.2. Molecular Identification of Ornithodoros Ticks and Detection of Borrelia *spp*. and Anaplasma *spp*. by Using qPCR in O. sonrai

Morphologically, all 253 ticks collected belonged to the species *O. sonrai*. A random selection of 18 *O. sonrai* tick specimens was performed for molecular confirmation. Using the tick *16S* gene, we were able to amplify the DNA of 33.33% (6/18) of these ticks. These six specimens were successfully sequenced. The blast results confirmed the identity of *O. sonrai*, with identification percentages ranging from 98.39% to 99.75% (Table 1). Of 253 tick DNA extracts tested using qPCR, one was found positive for *B. crocidurae* infection, leading to a prevalence of 0.39%. This single tick found infected was collected in a rodent burrow examined inside a store where food goods are sold. The results of the qPCR and the standard PCR of *Anaplasma* spp. detected in *O. sonrai* are detailed in Table 2. All ticks studied were negative for *Bartonella* spp., *C. burnetii* and *Rickettsia* spp.

### 3.3. Detection of B. crocidurae Infection in Rodents

Among the 150 DNA samples from rodent brain and spleen tissues extracted separately and tested with qPCR, 18 *Mus musculus domesticus* were found to be infected with *Borrelia* spp., which were later confirmed to be *B. crocidurae* using qPCR for the *B. crocidurae*-specific *glp*Q gene, with a prevalence of 12% (18/150) and a mean Ct of 28.81 (Table 3). Then, we were able to amplify 69% (9/13) of the samples identified as positive for *B. crocidurae* (CP003426.1/JX292897.1) with standard PCR and sequencing (Table 3). The other 31% (4/13) of samples found positive for *Borrelia* spp. according to the qPCR results could not be amplified using standard PCR, which may be due to PCR inhibitors [23]. All tested individuals of *A. niloticus* were negative for *Borrelia* spp.

### 3.4. Detection of Anaplasma *spp*. in Rodents

In *Mus musculus domesticus*, the qPCR results revealed 10.16% (6/59) and 6.78% (4/59) positive samples for *Anaplasma* spp. in the spleen and brain tissue of rodents, respectively. Standard PCR and sequencing were performed for the positive samples obtained from the qPCR. Blast showed that 10% (1/10) were identified as *Ehrlichia* sp. (Table 4). No rodent was found to be positive for *Bartonella* spp., *C. burnetii* or *Rickettsia* spp.

### 3.5. Phylogenetic Analysis

The phylogenetic analysis based on the Anaplasmataceae *23S* rRNA gene shows that our *Anaplasma* sp. is situated between *Anaplasma platys* (Dumler, Barbet, Bekker, Dasch, Palmer, Ray, Rikihisa, Rurangirwa, 2001) (M021425.1) and *Anaplasma ovis* (Lestoquard, 1924) (MT408585.1) (Figure 2). Using the Anaplasmataceae *16S* rRNA gene, it forms a separate clade of uncultured *Anaplasma* sp. (MK041546.1) (Figure 3).

Concerning *Ehrlichia* sp., its position on the tree was identical to that of Candidatus *Ehrlichia* sp. (MK484067.1) (Figure 2) and uncultured *Ehrlichia* sp. (MK041545.1) by targeting the Anaplasmataceae *23S* rRNA and Anaplasmataceae *16S* rRNA genes, respectively (Figure 3). Finally, with the *Ehrlichia groEL* gene, a distinct line between uncultured *Ehrlichia* sp. (MG385128.1) and *Ehrlichia ruminantium* (Dumler, Barbet, Bekker, Dasch, Palmer, Ray, Rikihisa, Rurangirwa, 2001) was formed (U13638.1) (Figure 4).

## 4. Discussion

Our study confirms the success of *Mus musculus domesticus* as an invasive species in rural areas of northern Senegal [12]. In November 2016, this species represented 85% of the small rodents captured in Dodel [24], while this proportion was approximately 98% in 2018 and 2021 (this study). The difference in relative abundance observed between the two years of sampling may be indicative of an increase in the population size of the species in this locality, a hypothesis that will have to be verified by performing further sampling. This expansion of house mice in rural areas of inner Senegal at the expense of native species has been shown to be associated with gastrointestinal helminth patterns, some of which likely favor this invasive success [24].

Using molecular tools, the present results also reveal a respective prevalence of *B. crocidurae* infection in one *O. sonrai* tick and 18 *Mus musculus domesticus* of 0.39% (1/253) and 12% (18/150), associated with an infestation rate of rodent burrows by ticks inside dwellings of 22.1% (23/104). Interestingly, the only *O. sonrai* tick found to be infected was from a burrow examined inside a shopkeeper’s store that hosted this single individual. The rate of *O. sonrai* ticks found to be infected with *B. crocidurae* was quite low compared to previous epidemiological studies conducted further south in western Senegal (the localities of Niakhar and Dielmo-Ndiop, especially) [5,7]. Although the abundance of *O. sonrai* ticks in some Dodel dwellings was high, the average value of 22.1% of burrows occupied by ticks appears below those recorded elsewhere in Senegal, with infestation percentages of 32%, 37% and 41% in the three regions of the country [5].

In small mammals, the prevalence of *Borrelia* infection ranges from 1.7% to 27.8% in West Africa [7]. In regions known to be endemic to TBRF in Senegal, such as Dielmo, Dakar, Thies and Richard-Toll, the prevalence of *Borrelia* infection was in the range of 13.8%-50% in two species of small mammals (*A. niloticus* and *Mus musculus domesticus*) [7,25,26,27]. The house mouse, acting as a natural reservoir of *Borrelia*, is a strictly commensal species in Senegal [13], and its abundant presence in small rodent communities, which maintains infection rates, makes it a risk factor for TBRF transmission [7]. Our survey reveals that *B. crocidurae* infection was more prevalent in house mice than in *O. sonrai* ticks. Indeed, the rate of *B. crocidurae* infection in *O. sonrai* ticks was particularly low. One may hypothesize that the contact between reservoir hosts and vector ticks may be rare, as *O. sonrai* ticks are essentially endophilic, while house mice often hide in closets, stacks of items in storerooms or other hiding places rather than in burrows.

The order *Rickettsiales* currently includes the family Anaplasmataceae [28], which is made up of the following genera: *Wolbachia*, *Ehrlichia*, *Anaplasma*, *Neorickettsia* and *Aegyptianella* [9]. The *Anaplasma* genus comprises Gram-negative intracellular bacteria [29]. These organisms cause anaplasmoses that affect human and animal health and are often transmissible by ticks [2,16,30]. In this work, we also found a potential new species of *Ehrlichia* in brain and spleen tissues of house mice from northern Senegal (Table 4). This result corroborates that of Dahmana et al. [9], who detected, among several other pathogens (including *B. crocidurae*), a potential new Candidatus “*Ehrlichia senegalensis*” in splenic tissue of rodents from the same area.

In Senegal, previous studies have reported the presence of Anaplasmataceae, with a high prevalence in small rodents [31,32]. In our study, we were able to amplify two potential new species, namely *Anaplasma* sp. and *Ehrlichia* sp., in *O. sonrai* ticks. In the phylogenetic trees, our potential new species of *Anaplasma* forms a separate cluster from other *Anaplasma* spp. (Figure 2 and Figure 3), which means that it does not conform to any previously recognized species, thus potentially representing a new species. Concerning our amplicon of *Ehrlichia* sp., based on the two genes Anaplasmataceae rRNA *23S* and Anaplasmataceae rRNA *16S*, our sequence was 100% identical with Candidatus *Ehrlichia* sp. and uncultured *Ehrlichia* sp., respectively. The latter represents a potential new species that has not yet been described (Figure 2 and Figure 3) [9]. With the *Ehrlichia groEL* gene, our amplicon forms a distinct cluster on the tree that could not be attributed to already recognized species (Figure 3). The detection of Anaplasmataceae in *O. sonrai* could be the consequence of a bacterial engorgement in ticks following a mixed infestation. This hypothesis has already been proposed for other tick species [33,34]. The vector competence of *O. sonrai* to transmit Anaplasmataceae cannot yet be confirmed, but there is a risk that it could be potentially a vector of this bacterium. Further studies and experimentation are needed to better understand the epidemiological relationship between Anaplasmataceae and these soft ticks.

## 5. Conclusions

This study contributes to the identification of an expanded repertoire of zoonotic bacteria hosted in *Mus musculus domesticus* house mice and *O. sonrai* ticks in northern Senegal. *Ornithodoros sonrai* ticks may be a potential vector of Anaplasmataceae. Further in-depth investigation and experimentation should be conducted to understand the potential epidemiological role of these ticks and their impact on human and animal health.

## Figures and Tables

**Figure 1 microorganisms-10-02367-f001:**
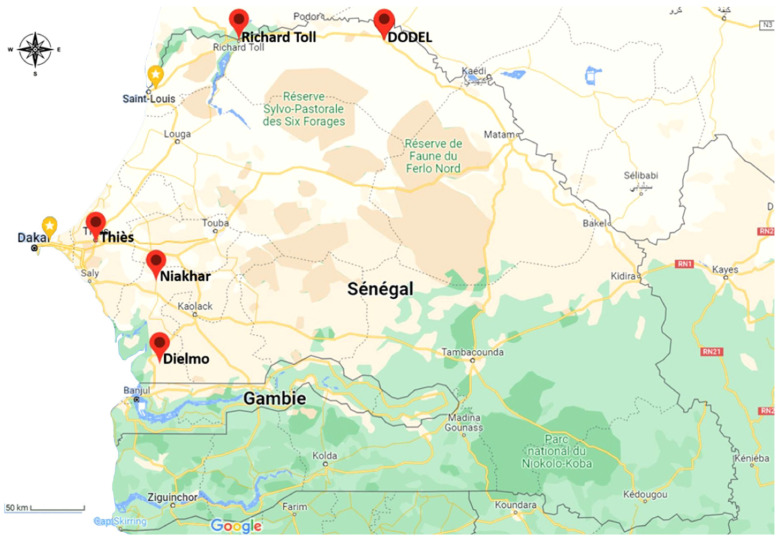
Location of the Dodel commune in the Saint-Louis region in Senegal.

**Figure 2 microorganisms-10-02367-f002:**
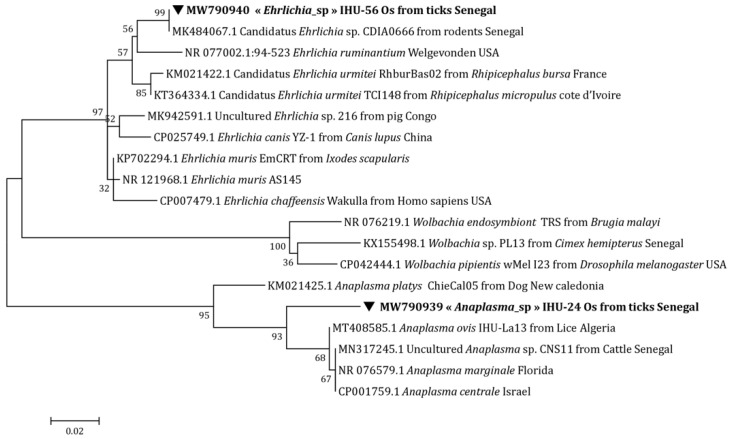
Anaplasmataceae maximum-likelihood phylogenetic tree targeting the partial 513-bp *23S* gene.

**Figure 3 microorganisms-10-02367-f003:**
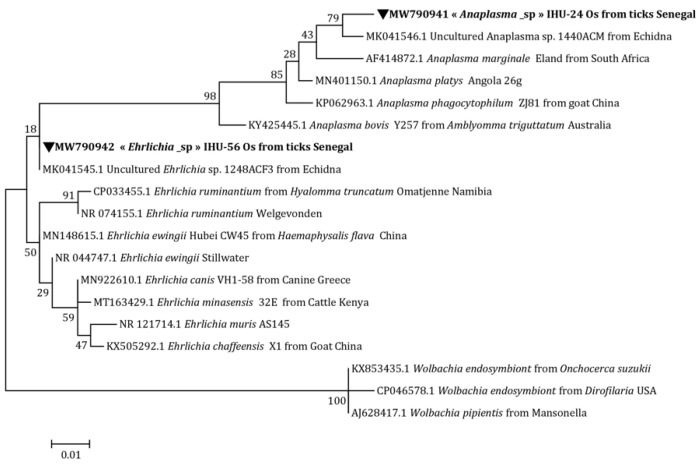
Anaplasmataceae maximum-likelihood phylogenetic tree targeting the partial 345 bp *16S* gene.

**Figure 4 microorganisms-10-02367-f004:**
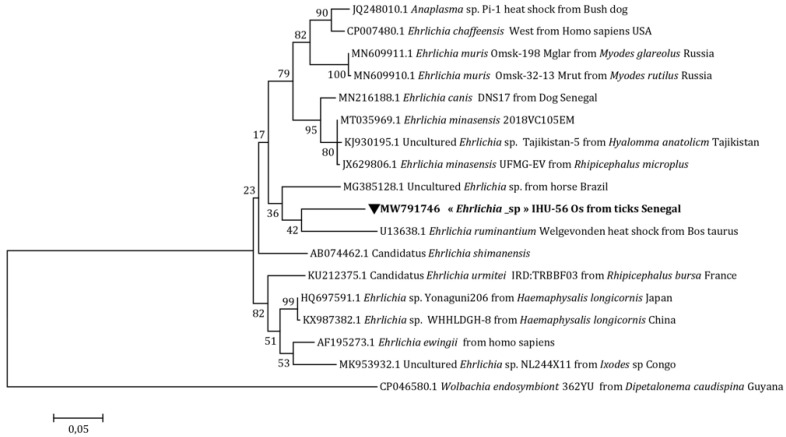
*Ehrlichia* spp. maximum-likelihood phylogenetic tree targeting the partial 633-bp *groEL* gene.

**Table 1 microorganisms-10-02367-t001:** Blast analysis results for molecular identification of *Ornithodoros sonrai* ticks collected in Dodel, Senegal.

Individual Samples of *Ornithodoros* Ticks	PCR Standard *16S* of *O. sonrai* Ticks	Cover	Per. Ident	Accession Number of the Reference Sequences	Identification
*Ornithodoros sonrai*	(+)	99%	99.75%	KP644213.1	*O. sonrai*
*O. sonrai*	(+)	99%	99.75%	KP644213.1	*O. sonrai*
*O. sonrai*	(+)	100%	98.87%	KP644213.1	*O. sonrai*
*O. sonrai*	(+)	95%	99.30%	KP644213.1	*O. sonrai*
*O. sonrai*	(+)	95%	98.39%	KP644213.1	*O. sonrai*
*O. sonrai*	(+)	95%	99.53%	KP644213.1	*O. sonrai*
Total	6/6	/	/	/	*O. sonrai*

**Table 2 microorganisms-10-02367-t002:** Results of qPCR, standard PCR and BLAST from *O. sonrai* tested by targeting primers *Anaplasma* spp. *23S* rRNA, Anaplasmataceae *16S* rRNA and *Ehrlichia groEL*.

Individual Samples of *Ornithodoros* Ticks	qPCR Primer	qPCR (Ct)	Bacteria Detected in *O. sonrai* Ticks Using qPCR	PCR Standard Primer	Molecular Identification Using Blast	Percent Identity %	Query Cover %	Accession Number of the Reference Sequences
*Ornithodoros sonrai*	Anaplasmataceae *23S* rRNA gene	7.23	*Anaplasmataceae* spp.	Anaplasmataceae *23S* rRNA gene	Candidatus *Ehrlichia* sp.	100	95	MK484067.1
		/	*Anaplasmataceae* spp.	Anaplasmataceae *16S* rRNA gene	Uncultured *Ehrlichia* sp.	100	100	MK041545.1
		/	*Anaplasmataceae* spp.	*Ehrlichia groEL*	*Ehrlichia* sp.	90.50	100	HQ697591.1
*O. sonrai*		20.68	*Anaplasmataceae* spp.	Anaplasmataceae *23S* rRNA gene	*Anaplasma ovis*	94.97	94	MT408585.1
		/	*Anaplasmataceae* spp.	Anaplasmataceae *16S* rRNA gene	Uncultured *Anaplasma* sp.	98.77	100	MK041546.1
*O. sonrai*		9.75	*Anaplasmataceae* spp.	/	/	/	/	/
*O. sonrai*		11.12	*Anaplasmataceae* spp.	/	/	/	/	/
*O. sonrai*		29.93	*Anaplasmataceae* spp.	/	/	/	/	/
*O. sonrai*		14.07	*Anaplasmataceae* spp.	/	/	/	/	/
*O. sonrai*		7.05	*Anaplasmataceae* spp.	/	/	/	/	/
*O. sonrai*		21.78	*Anaplasmataceae* spp.	/	/	/	/	/
*O. sonrai*		20.69	*Anaplasmataceae* spp.	/	/	/	/	/
Total	Ct m = 15.81			/	/	/	/	/

Ct m = The threshold value of an average cycle.

**Table 3 microorganisms-10-02367-t003:** Results of *Borrelia* spp. detection in brain and spleen tissue from rodents caught in Dodel using qPCR and standard PCR.

Rodent Tissue Studied	Ct (qPCR)	PCR Standard	Cover	Per. Ident	Accession Number of the Reference Sequences	Identification
*Mus musculus domesticus* ^b^	30.29	(+)	99%	97.92%	CP003426.1	*B. crocidurae*
*M. m. domesticus* ^b^	33.55	/	/	/	/	/
*M. m. domesticus* ^b^	10.35	/	/	/	/	/
*M. m. domesticus* ^b^	23.13	(+)	100%	99.93%	CP003426.1	*B. crocidurae*
*M. m. domesticus* ^b^	28	(+)	100%	95.35%	JX292897.1	*B. crocidurae*
*M. m. domesticus* ^b^	28.11	(+)	99%	98.95%	CP003426.1	*B. crocidurae*
*M. m. domesticus* ^b^	27.16	(+)	99%	98.63	CP003426.1	*B. crocidurae*
*M. m. domesticus* ^b^	27.96	(+)	100%	99.41%	CP003426.1	*B. crocidurae*
*M. m. domesticus* ^b^	29.28	(+)	99%	98.44%	CP003426.1	*B. crocidurae*
*M. m. domesticus* ^b^	24.74	(+)	99%	99.40%	CP003426.1	*B. crocidurae*
*M. m. domesticus* ^s^	35.81	/	/	/	/	/
*M. m. domesticus* ^s^	36.34	/	/	/	/	/
*M. m. domesticus* ^s^	30.14	(+)	96%	99,85%	CP003426.1	*B. crocidurae*
*M. m. domesticus* ^b^	33.47		/	/		/
*M. m. domesticus* ^b^	29.63		/	/	/	/
*M. m. domesticus* ^b^	24.81		/	/	/	/
*M. m. domesticus* ^b^	32.19		/	/	/	/
*M. m. domesticus* ^b^	33.72		/	/	/	/
Total	28.81	69% (9/13)	/	/	/	/

^b^ = brain tissue; ^s^ = spleen tissue.

**Table 4 microorganisms-10-02367-t004:** Results of *Anaplasma* spp. detection in brain and spleen tissue from mice caught in Dodel using qPCR and standard PCR.

Small Rodent Tissue Studied	Ct (qPCR)	PCR Standard	Cover	Per. Ident	Accession Number	Identification
*M. m. domesticus* ^s^	25.47	/	/	/	/	/
*M. m. domesticu* ^s^	26.13	/	/	/	/	/
*M. m. domesticu* ^b^	29.21	/	/	/	/	/
*M. m. domesticu* ^s^	17.95	/	/	/	/	/
*M. m. domesticus* ^b^	20.18	/	/	/	/	/
*M. m. domesticus* ^s^	24.78	/	/	/	/	/
*M. m. domesticus* ^s^	22.49	(+)	92%	100	MK484067.1	Candidatus *Ehrlichia* sp.
*M. m. domesticus* ^s^	27.57	/	/	/	/	/
*M. m. domesticus* ^b^	18.42	/	/	/	/	/
*M. m. domesticus* ^b^	20.45	/	/	/	/	/
Total	23.26	10% (1/10)	/	/	/	/

^b^ = brain tissue; ^s^ = spleen tissue.

## Data Availability

Summary data on small mammals can be publicly accessed via http://BPM-CBGP.science (accessed on 11 October 2022).

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
