# Peer review of "Pathogen Detection in Ornithodoros sonrai Ticks and Invasive House Mice Mus musculus domesticus in Senegal"

_microorganisms, 2022, doi:10.3390/microorganisms10122367_

Round 1

Reviewer 1 Report

The authors describe the occurrence of selected pathogens in Ornitodoros sonrai ticks and in house mouse Mus musculus from Africa/Senegal. Undoubtedly, these are important data from the point of view of humans and pathogen transmission.

I am of an opinion that the article fits into scope of Microorganisms and could be published after major corrections.

Comments:

1. Title:

pathogens and not diseases were detected in ticks and mammals.

2. Introduction:

„Ticks are strict blood-sucking arthropods divided into two families, Ixodidae (about 700 hard ticks) and Argasidae (around 200 soft ticks) species „ – incorrect citation, there is no information on the number of Acari species in this article [1]; besides, some taxonomic work should be cited here. Ticks do not transmit disease, only pathogens; see also title.

3. Material and methods:

The authors write that “Cats were found in the majority of the houses sampled” – however, there is no comment here (maybe in the Discussion) as to whether this information about cats has anything to do with this work.

4. Sampling:

Please add the number of samples for the rodent burrows, domestic and peri-domestic crevices; for each separately.

“All specimens caught were identified according to morphological criteria as previously reported [5,7]” – incorrect citations, there is no information about identification of rodents in these articles ?! Please check the correctness of the other citations, because I can see that there are serious errors here.

5. Ornithodoros tick identification and DNA extraction:

“All Ornithodoros ticks were subjected to standard PCR and sequencing to confirm morphological identification using the 16S tick gene” – did this also apply to the ticks previously used for SEM?.

6. Screening for bacteria:

Please do not use "small rodent" only specific species names, or use only “rodents”. The note concern to Material and methods and Results.

„bacteria such as Anaplasma spp., Borrelia spp., Bartonella spp., C. burnetii and Rickettsia spp.” – the abbreviations "spp." are not needed here.

7. Results:

“Overall, 150 small rodents were captured (59 in 2018 and 91 in 2021) over 966 trapnights”; “They belonged to two species of the family Muridae; i.e., Mus musculus domesticus (147 individuals) and Arvicanthis niloticus (3 individuals)”; and “A total of 104 small rodent burrows were examined in dwellings and peri-domestic areas,”– this belong to Material and methods; see also comment 4.

Please explain what the "infestation rate" parameter means.

Please clearly state which ticks (rodent burrows, domestic and peri-domestic crevices) had pathogens; I do not see information where the infected ticks came from? This is important in the context of the possibility of transmission of pathogens from O. sonrai and M. musculus; the more that the M. musculus lives mainly in the area of human houses / rooms, and the O. sonrai in burrows.

8. Tables 1, 2:

Since only one species of tick was found, it makes no sense to write "species of O. ticks", otherwise these are not species but individual samples (6).

9. Table 3:

Title / first column – please delete „small”.

10. Discussion:

“Our study confirms the success of Mus musculus domesticus as an invasive species in rural areas of northern Senegal. In November 2016, this species represented 85% of the small rodents captured in Dodel [21], while this proportion was approximately 98% in 2018 and 2021 (this study). The difference in relative abundance observed between the two years of sampling may be indicative of an increase in its population in this locality, a hypothesis that will have to be checked by further sampling. This expansion of house mice in rural areas of inner Senegal at the expense of native species has been shown to be associated with gastro-intestinal helminth patterns, some of which likely favor this invasive success[21]” – it was not the purpose of this work after all; how the authors defined the success of Mus musculusas an invasive species in Senegal ?.

11. Please give complete scientific names with authors and dates. The more that, for example, there are two species of Arvicanthis niloticus with different status: Arvicanthis niloticus (Desmarest, 1822) – invalid and Arvicanthis niloticus (É. Geoffroy, 1803) – valid status. The comment applies to the entire text.

12. Please format the bibliography according to the Microorganisms MDPI.

Author Response

Reviewer #1: Dear editor

The authors describe the occurrence of selected pathogens in Ornitodoros sonrai ticks and in house mouse Mus musculus from Africa/Senegal. Undoubtedly, these are important data from the point of view of humans and pathogen transmission.

I am of an opinion that the article fits into scope of Microorganisms and could be published after major corrections.

Comments:

  1. Title:

pathogens and not diseases were detected in ticks and mammals.

 Author’s answer:

Thank you for your comment we have corrected the title on the lines 1-2.

“Pathogens Detection in Ornithodoros sonrai Ticks and the Invasive House Mice Mus musculus domesticus in Senegal ”.

  1. Introduction:

„Ticks are strict blood-sucking arthropods divided into two families, Ixodidae (about 700 hard ticks) and Argasidae (around 200 soft ticks) species „ – incorrect citation, there is no information on the number of Acari species in this article [1]; besides, some taxonomic work should be cited here. Ticks do not transmit disease, only pathogens; see also title.

Author’s answer:

Thank you for your comment, we have corrected and added references on lined 39-42. “The latter comprises the sub-families Argasinae and Ornithodorinae within the genus Ornithodoros [1,2]. Ticks can have a negative impact on an animal's health. They are the second group in the transmission of infectious pathogens to humans after mosquitoes [3]”.

  1. Material and methods:

The authors write that “Cats were found in the majority of the houses sampled” – however, there is no comment here (maybe in the Discussion) as to whether this information about cats has anything to do with this work.

Author’s answer:

Thank you for your remark, we deleted this sentence from the manuscript because it has no impact in this study it was just an observation in the field.

  1. Sampling:

Please add the number of samples for the rodent burrows, domestic and peri-domestic crevices; for each separately.

“All specimens caught were identified according to morphological criteria as previously reported [5,7]” – incorrect citations, there is no information about identification of rodents in these articles ?! Please check the correctness of the other citations, because I can see that there are serious errors here.

Author’s answer:

Thank you for your comments we have added the number of samples for rodent burrows, domestic and peri-domestic crevices; for each separately. And we have corrected the references on these lines 148-151 “ Overall, 150 rodents (59 in 2018 and 91 in 2021) were captured over 966 trap-nights, leading to an overall trapping success of 15.22%. They belonged to two species of the family Muridae, i.e. M. m. domesticus (147 individuals) and Arvicanthis niloticus (A. niloticus) (É. Geoffroy, 1803) (3 individuals)”. And on the line 155 “ A total of 104 rodent burrows were examined in dwellings and peri-domestic areas”. 

  1. Ornithodoros tick identification and DNA extraction:

“All Ornithodoros ticks were subjected to standard PCR and sequencing to confirm morphological identification using the 16S tick gene” – did this also apply to the ticks previously used for SEM?.

Author’s answer:

Thanks for your comments, morphological identification was confirmed for all ticks also on ticks used for SEM.

  1. Screening for bacteria:

Please do not use "small rodent" only specific species names, or use only “rodents”. The note concern to Material and methods and Results.

„bacteria such as Anaplasma spp., Borrelia spp., Bartonella spp., C. burnetii and Rickettsia spp.” – the abbreviations "spp." are not needed here.

Author’s answer:

Thank you for your comments, we used rodents for the material and method and results passages, so we deleted spp on the lines 121.

“bacteria such as Anaplasma, Borrelia, Bartonella, C. burnetii and Rickettsia”.

  1. Results:

“Overall, 150 small rodents were captured (59 in 2018 and 91 in 2021) over 966 trapnights”; “They belonged to two species of the family Muridae; i.e., Mus musculus domesticus (147 individuals) and Arvicanthis niloticus (3 individuals)”; and “A total of 104 small rodent burrows were examined in dwellings and peri-domestic areas,”– this belong to Material and methods; see also comment 4.

Please explain what the "infestation rate" parameter means.

Please clearly state which ticks (rodent burrows, domestic and peri-domestic crevices) had pathogens; I do not see information where the infected ticks came from? This is important in the context of the possibility of transmission of pathogens from O. sonrai and M. musculus; the more that the M. musculus lives mainly in the area of human houses / rooms, and the O. sonrai in burrows.

Author’s answer:

Thank you for your comment, we have added the information on these lines 148-151 “Overall, 150 rodents (59 in 2018 and 91 in 2021) were captured over 966 trap-nights, leading to an overall trapping success of 15.22%. They belonged to two species of the family Muridae, i.e. M. m. domesticus (147 individuals) and Arvicanthis niloticus (A. niloticus)  (É. Geoffroy, 1803) (3 individuals)”. And on the lines 155 “A total of 104 rodent burrows were examined in dwellings and peri-domestic areas”.

And we explained what the infestation rate means on the lines 156-157.

“An overall burrow infestation rate of 22.1% was recorded (23/104 rodent burrows were infested with Ornithodorosticks), leading to the collection of 253 specimens of ticks”.

and indicated the ticks that had the pathogens on these lines 166-167.

“This single tick found infected was collected in a rodent burrow examined inside a store where food goods are sold”.

  1. Tables 1, 2:

Since only one species of tick was found, it makes no sense to write "species of O. ticks", otherwise these are not species but individual samples (6).

Author’s answer:

Thank you for your remark and we apologize for this error, these are individual samples we have corrected on table 1 and 2 lines 385 and 388 respectively.

  1. Table 3:

Title / first column – please delete „small”.

Author’s answer:

Thank you for your remark, we have removed small from the Title / first column table 3 in the line 388.

  1. Discussion:

“Our study confirms the success of Mus musculus domesticus as an invasive species in rural areas of northern Senegal. In November 2016, this species represented 85% of the small rodents captured in Dodel [21], while this proportion was approximately 98% in 2018 and 2021 (this study). The difference in relative abundance observed between the two years of sampling may be indicative of an increase in its population in this locality, a hypothesis that will have to be checked by further sampling. This expansion of house mice in rural areas of inner Senegal at the expense of native species has been shown to be associated with gastro-intestinal helminth patterns, some of which likely favor this invasive success [21]” – it was not the purpose of this work after all; how the authors defined the success of Mus musculus as an invasive species in Senegal ?.

Author’s answer:

Even if the invasion success was not the main purpose of this work, it seems important to us to mention it as an element of context enabling to highlight the potentially increasing risk of transmission of borreliosis and other zoonotic diseases to humans associated with i) the high prevalence of Borrelia crocidurae in Mus musculus and ii) the success of invasion of this species in Senegal, clearly showed by Dalecky et al. (2015) [ref.12] and confirmed by Diagne et al. (2021) [ref. 24] as well as in our present study in Dodel. In all these studies, invasion success of the house mouse is measured by both the eastward extension of its distribution in the country and its high population size in most of the villages (including Dodel) sampled.

  1. Please give complete scientific names with authors and dates. The more that, for example, there are two species of Arvicanthis niloticus with different status: Arvicanthis niloticus (Desmarest, 1822) – invalid and Arvicanthis niloticus (É. Geoffroy, 1803) – valid status. The comment applies to the entire text.

Author’s answer:

Thank you for your comment, we have given full scientific names with authors and dates throughout the text.

  1. Please format the bibliography according to the Microorganisms MDPI.

Author’s answer:

Thank you for your comment we have formatted the bibliography according to the MDPI Microorganisms.

Reviewer 2 Report

Dear authors,

The study is well executed and will help to improve knowledge on the prevalence and distribution of tick-borne relapsing fever bacteria in relation to reservoir hosts and vector ticks, which could be useful in applying further surveillance plans.

Suggestions and comments of the reviewer:

Abstract:

1)    Write the abbreviation of the species after the full name the first time it appears: Ornithodoros sonrai (O. sonrai) ticks’; ‘...and in house mice, Mus musculus domesticus (M. m. domesticus), as…’.  Also, use this format for the name of the bacterium in the Introduction chapter, and then always use the abbreviations.  

Introduction:

2)    Rephrase: They are the second of infectious pathogens for humans after mosquitoes’, for exemple: They are the second in the transmission of infectious pathogens to humans after mosquitoes.

  3)    ‘..In Senegal, TBRF is a major cause of morbidity, although..’ Change ‘morbidity’ with ‘diseases’.

4)    The symbol % is reported to times: 72.72%%

5)    In the sentence: ‘A large-scale screening in North and West Africa detected the presence of B. crocidurae, B. hispanica, and B. merionesi in O. sonrai, O.marocanus group, O. merionesi and O. costalis ticks, respectively’,  you reported three borrelelia species and four tick species. Can you better explain the term, respectively?

6)    In the sentence: ‘We also sought to identify an expanded repertoire of infectious diseases in order to..’,  change the term diseases with agents .  

Materials and Methods   Chapter 2.2 Sampling

7)    You have stated that rodents were captured alive and processed according to live animal capture guidelines, but the brain, spleen, and other organs were used for research. You need to explain the culling procedure and state that it complies with the legislation. I think you need the approval of an ethics committee, and the approval document number should be given.  

Chapter 2.4 Screening for bacteria

  8)    Add the verb in the sentence: ‘Samples positive with the Anaplasmataceae 23S rRNA gene were ……by..

9)    I suggest moving the sentence: ‘The accession numbers assigned by GenBank …. finally for the gene Ehrlichia groEL (Ehrlichia sp. (MW791746)).’ at the end of Chapter 2.4, before the sentence on phylogenetic analyses.

10) The tissue used for extraction is not well specified, and the results section does not specify which tissues tested positive for infectious agents. Was a mix of brain and spleen used for extraction? Were both tissues extracted separately for each rodent, or sometimes the brain and sometimes the spleen? Please specify in Materials and Methods and consequently in the Results section.  

Results

Chapter 3.2  

11)  Since only 33.33% of the tick DNA was amplified, is it possible that there were inhibitors of real-time reaction?

12) In Table 2, check the ‘E’ of the Ehrlichia name and write it in italics.  

  Chapter 3.3

 13) The reviewer suggests changing the sentence: ‘18 M. m. domesticus were found to be infected with Borrelia spp., later confirmed to be B. crocidurae, with a prevalence of 12% (18/150) and a mean Ct of 28.76.'.

14) Table 2: change the the RT-PCR in the title of the columns to qPCR.

15) In the opinion of reviewer, the results on the positive and unconfirmed B. crocidurae samples should be better explained and commented on. Table 3 shows that, other than the four unconfirmed B. crocidurae samples, other qPCR-positive samples were not amplified by standard PCR (only 9 positive PCR samples are reported and not 14). Can you explain these results?

16) The accession numbers of the B. crucidurae sequences should be reported.

17) Specify in the Tables 1, 2, and 3 that the accession number is of reference sequences.  

Chapter 3.4  

18) Please, correct the first sentence: ‘…Anaplasma spp. Spleen and brain tissue, respectively’.  

Chapter 3.5  

19) It would be interesting to also have the phylogenetic analysis of Borrelia.  

Discussion  

20)  In the sentence: ‘The detection of Anaplasmataceae in O. sonrai could be the consequence of a bacterial engorgement following mixed infestation’, do you mean tick engorgement?

Author Response

Reviewer #2

The study is well executed and will help to improve knowledge on the prevalence and distribution of tick-borne relapsing fever bacteria in relation to reservoir hosts and vector ticks, which could be useful in applying further surveillance plans.

Abstract:

1)    Write the abbreviation of the species after the full name the first time it appears: ‘Ornithodoros sonrai (O. sonrai) ticks’; ‘...and in house mice, Mus musculus domesticus (M. m. domesticus), as…’.  Also, use this format for the name of the bacterium in the Introduction chapter, and then always use the abbreviations.  

Author’s answer:

Thank you for your comment we have corrected this on the whole manuscript.

Introduction:

2)    Rephrase: They are the second of infectious pathogens for humans after mosquitoes’, for exemple: They are the second in the transmission of infectious pathogens to humans after mosquitoes.

Author’s answer:

Thank you for your comment, we have corrected this sentence on the lines 41-42.

“They represent the second group for the transmission of infectious pathogens to humans after mosquitoes”.

3)    ‘..In Senegal, TBRF is a major cause of morbidity, although..’ Change ‘morbidity’ with ‘diseases’.

Author’s answer:

Thank you for your comment, we have replaced morbidity with the word disease on the line 54. “In Senegal, TBRF is a major cause of diseases”.

4)    The symbol % is reported to times: 72.72%%

Author’s answer:

Thank you for your comment, we apologize for this error we have removed the second percentage on the line 57.

“with a prevalence ranging from 4.54% to 72.72%.”

5)    In the sentence: ‘A large-scale screening in North and West Africa detected the presence of B. crocidurae, B. hispanica, and B. merionesi in O. sonrai, O.marocanus group, O. merionesi and O. costalis ticks, respectively’,  you reported three borrelelia species and four tick species. Can you better explain the term, respectively?

Author’s answer:

Thank you for your comment, we have corrected this sentence on the lines 58-62.

“A large-scale screening in North and West Africa detected the presence of B. crocidurae, in O. sonrai ticks; B. hispanicain Ornithodoros marocanus (Velu, 1919), Ornithodoros occidentalis and Ornithodoros kairouanensis (Trape et al., 2013); and Borrelia merionesi in Ornithodoros merionesi (Trape et al., 2013 ) and Ornithodoros costalis (Diatta et al., 2013) [5]”.

6)    In the sentence: ‘We also sought to identify an expanded repertoire of infectious diseases in order to..’,  change the term diseases with agents .  

Author’s answer:

Thank you for your comment, we have replaced the term diseases with agents on the line 71.  “expanded repertoire of infectious agents in order”.

Materials and Methods   Chapter 2.2 Sampling

7)    You have stated that rodents were captured alive and processed according to live animal capture guidelines, but the brain, spleen, and other organs were used for research. You need to explain the culling procedure and state that it complies with the legislation. I think you need the approval of an ethics committee, and the approval document number should be given.  

Author’s answer:

Thank you for your comment, We have added information on culling procedure (by cervical dislocation, as recommended by the Animal Care and Use Committee [2011]), and precised that our protocols were validated by the Regional Head of the Veterinary Service, Hérault, France under agreement D-34-169-003 that covers all experimental activities on rodents led by the CBGP team (incl. L. Granjon and J. Le Fur, among others) on these lines 92-102.

“Rodents were euthanized by cervical dislocation and handled in accordance with the relevant requirements of Senegalese legislation and live animal capture guidelines of the American Society of Mammalogists (Animal Care and Use Committee 2011). Body measurements were taken before autopsy and reproductive status was noted. Brain and spleen tissue as well as digestive tracts were preserved in ethanol 95% for further analyses. All protocols used here were conducted following official regulations (Centre de Biologie pour la Gestion des Populations (CBGP): Agrément pour l’utilisation d’animaux à des fins scientifiques D-34-169-003) of the relevant institutional committee (Regional Head of the Veterinary Service, Hérault, France). All transfer and conservation procedures were performed in accordance with current Senegalese and French legislation”.

Chapter 2.4 Screening for bacteria

  8)    Add the verb in the sentence: ‘Samples positive with the Anaplasmataceae 23S rRNA gene were ……by..

Author’s answer:

Thank you for your comment, we added the verb tested to the sentence on the lines 128-129. “Samples positive with the Anaplasmataceae 23S rRNA gene were tested by an additional”.

9)    I suggest moving the sentence: ‘The accession numbers assigned by GenBank …. finally for the gene Ehrlichia groEL (Ehrlichia sp. (MW791746)).’ at the end of Chapter 2.4, before the sentence on phylogenetic analyses.

Author’s answer:

Thank you for your comment, this paragraph has been moved to the end of the chapter on these lines 141-145.

“The accession numbers assigned by GenBank to all our sequences are listed as follows: for the Anaplasmataceae 23SrRNA gene (Anaplasma sp. (MW790939) and Ehrlichia sp. (MW790940)); for the Anaplasmataceae 16S rRNA gene (Anaplasma sp. (MW790941) and Ehrlichia sp. (MW790942)) and finally for the gene Ehrlichia groEL (Ehrlichia sp. (MW791746))”.

10) The tissue used for extraction is not well specified, and the results section does not specify which tissues tested positive for infectious agents. Was a mix of brain and spleen used for extraction? Were both tissues extracted separately for each rodent, or sometimes the brain and sometimes the spleen? Please specify in Materials and Methods and consequently in the Results section.  

Author’s answer:

Thank you for your comment, we have added this in the materials, methods and results section. on these lines 112-113  “DNA extraction was performed on a small portion of spleen and brain tissue separately for each rodent”.

And these lines 171-172 “Among the 150 DNA from rodents brain and spleen tissue extracted separately and tested by qPCR”.

concerning tissues positive for infectious agents have been specified in Tables 3 and 4.

Results

Chapter 3.2  

11)  Since only 33.33% of the tick DNA was amplified, is it possible that there were inhibitors of real-time reaction?

Author’s answer:

Thank you for your comment, yes, it is also believed to be due to the presence of PCR inhibitors which prevented DNA amplification showed by Bessetti et al [ref. 23].

12) In Table 2, check the ‘E’ of the Ehrlichia name and write it in italics.  

Author’s answer:

Thank you for your remark, we have corrected on table 2 on the line 383.

Chapter 3.3

 13) The reviewer suggests changing the sentence: ‘18 M. m. domesticus were found to be infected with Borrelia spp., later confirmed to be B. crocidurae, with a prevalence of 12% (18/150) and a mean Ct of 28.76.'.

Author’s answer:

Thank you for your comment, we changed this sentence on lines 172-174 of the manuscript.

“18 M. m. domesticus were found to be infected with Borrelia spp., which were later confirmed to be B. crocidurae using qPCR for B. crocidurae specific glpQ gene, with a prevalence of 12% (18/150) and a mean Ct of 28.81 (Table 3)”.

14) Table 2: change the the RT-PCR in the title of the columns to qPCR.

Author’s answer:

Thank you for your comment we have corrected on table 2.

15) In the opinion of reviewer, the results on the positive and unconfirmed B. crocidurae samples should be better explained and commented on. Table 3 shows that, other than the four unconfirmed B. crocidurae samples, other qPCR-positive samples were not amplified by standard PCR (only 9 positive PCR samples are reported and not 14). Can you explain these results?

Author’s answer:

Thank you for your comment, Please note that the last 5 infected Mus musculus domesticus were not blasted, so no DNA sequences were available as the positive DNAs were lost. It was inadvertent that the columns of “Accession Number of the reference sequences and Identification” were filled in, and the corresponding data mentioned in the table 3 are now deleted. We have explained these results on lines 174-178. “Then, we were able to amplify by standard PCR and sequencing 69% (9/13) of the samples identified as positive for B. crocidurae (CP003426.1 / JX292897.1) (Table 3). The other 31% (4/13) of samples positive for Borrelia spp. by qPCR could not be amplified by standard PCR, which may be due to PCR inhibitors [23]. All individuals of A. niloticus tested were negative for Borrelia spp”.

16) The accession numbers of the B. crucidurae sequences should be reported.

Author’s answer:

Thank you for your comment, we have indicated the accession numbers of the sequences of B. crucidurae on the line 176. “B. crocidurae (CP003426.1 / JX292897.1) (Table 3)”.

17) Specify in the Tables 1, 2, and 3 that the accession number is of reference sequences.  

Author’s answer:

Thank you for your comment, we have specified in Tables 1, 2 and 3 that the accession number is that of the reference sequences.

Chapter 3.4  

18) Please, correct the first sentence: ‘…Anaplasma spp. Spleen and brain tissue, respectively’.  

Author’s answer:

Thank you for your comment, we corrected the sentence on these lines 180-181.

“positive samples for Anaplasma spp. in the spleen and brain tissue of rodents, respectively”.

Chapter 3.5  

19) It would be interesting to also have the phylogenetic analysis of Borrelia.  

Author’s answer:

Thank you for your comment, indeed the phylogenetic tree for Borrelia was not made, since the positive samples for Borrelia Crocidurae were confirmed by two methods: qPCR using two genes and standard PCR and sequencing.

Discussion  

20)  In the sentence: ‘The detection of Anaplasmataceae in O. sonrai could be the consequence of a bacterial engorgement following mixed infestation’, do you mean tick engorgement?

Author’s answer:

Thanks for your comment, yes indeed we meant the ticks that are engorged, we corrected the sentence on the line 245-247.

”The detection of Anaplasmataceae in O. sonrai could be the consequence of a bacterial engorgement in ticks following a mixed infestation”.

Round 2

Reviewer 1 Report

I have a few more comments that need to be corrected.

4. Admittedly, I do not see that the authors have completed the data on the number of samples for the rodent burrows, domestic and peri-domestic crevices; for each separately. But I understand that the authors have only the total number without dividing it into rodent burrows, domestic and peri-domestic crevices.

7. As I wrote this text „Overall, 150 small rodents were captured (59 in 2018 and 91 in 2021) over 966 trapnights”; “They belonged to two species of the family Muridae; i.e., Mus musculus domesticus (147 individuals) and Arvicanthis niloticus (3 individuals)”; and “A total of 104 small rodent burrows were examined in dwellings and peri-domestic areas,”– belong to Material and methods. The work is about pathogens - and this is the Result and rodents were the Material.

Unfortunately, I don't see an explanation of the term "infestation rate."

11. Unfortunately, the authors did not complete the scientific names of the cited species, and the ones they completed are wrong !

For example, the correct notation for Mus musculus domesticus is Mus musculus domesticus Schwarz and Schwarz, 1943 - without parentheses !!!, this type of notation is regulated by the appropriate nomenclature code, in this case the International Code of Zoological Nomenclature. Sometimes authors with the date are in parentheses, and sometimes not - these have meaning !!! These designations mean something, for example, if the original species was placed in another genus - then the authors are in parentheses.

Should be : Ornithodoros marocanus Velu, 1919

Ornithodoros occidentalis Trape, Diatta, Durand & Renaud, 2013 etc, etc, etc.

Please do not use the notation "Mus musculus domesticus (M. m. domesticus)" - the abbreviation given in parentheses is not necessary and has no justification.

Author Response

Authors’ response - Manuscript ID: microorganisms-1995369

Full title: Pathogens Detection in Ornithodoros sonrai Ticks and the Invasive House Mice Mus musculus domesticus in Senegal

Corresponding author: Philippe PAROLA

Journal: Microorganisms

Please find the authors’ answer to the reviewer:

Reviewer 1

Comments and Suggestions for Authors

I have a few more comments that need to be corrected.

  1. Admittedly, I do not see that the authors have completed the data on the number of samples for the rodent burrows, domestic and peri-domestic crevices; for each separately. But I understand that the authors have only the total number without dividing it into rodent burrows, domestic and peri-domestic crevices.

Author’s answer:

Thank you for your comment, please note that there were only two crevices examined and neither contained the tick Ornithodoros sonrai. Indeed, it would have been interesting to give separate indications of the number of rodent burrows and crevices found infested with ticks if any crevices contained O. sonrai ticks. Since only rodent burrows were infested, we felt it necessary to present the tick sampling results without specifying the number of burrows and crevices.

  1. As I wrote this text „Overall, 150 small rodents were captured (59 in 2018 and 91 in 2021) over 966 trapnights”; “They belonged to two species of the family Muridae; i.e., Mus musculus domesticus (147 individuals) and Arvicanthis niloticus (3 individuals)”; and “A total of 104 small rodent burrows were examined in dwellings and peri-domestic areas,”– belong to Material and methodsThe work is about pathogens - and this is the Result and rodents were the Material.

Author’s answer:

Thank you for your comment, we moved these passages in materials and methods on the lines 105-108 And 112 -113.

“Overall, 150 rodents (59 in 2018 and 91 in 2021) were captured over 966 trap-nights, leading to an overall trapping success of 15.22%. They belonged to two species of the family Muridae, i.e. M. m. domesticus (147 individuals) and Arvicanthis niloticus  (É. Geoffroy, 1803) (3 individuals)”.

 “A total of 104 rodent burrows were examined in dwellings and peri-domestic areas”.

Unfortunately, I don't see an explanation of the term "infestation rate."

Thank you for your comment the infestation rate (22.1%) is calculated as the ratio of burrows or crevices containing ticks to all burrows or crevices studied." the explanation has been given in lines 113-115; 160- 161 and 212-214.

“The rodent burrow infestation rate was calculated as the ratio of the number of burrows harboring ticks to the total number of burrows sampled”.

 “Otherwise, an overall burrow infestation rate of 22.1% was recorded (23/104 rodent burrows were infested with O. sonrai ticks), leading to the collection of 253 specimens of ticks”.

“associated with an infestation rate of rodent burrows by ticks inside dwellings of 22.1% (23/104). Interestingly, the only O. sonrai tick found to be infected was from a burrow examined inside a shopkeeper's store that hosted this single individual”.

  1. Unfortunately, the authors did not complete the scientific names of the cited species, and the ones they completed are wrong !

For example, the correct notation for Mus musculus domesticus is Mus musculus domesticus Schwarz and Schwarz, 1943 - without parentheses !!!, this type of notation is regulated by the appropriate nomenclature code, in this case the International Code of Zoological Nomenclature. Sometimes authors with the date are in parentheses, and sometimes not - these have meaning !!! These designations mean something, for example, if the original species was placed in another genus - then the authors are in parentheses.

Should be : Ornithodoros marocanus Velu, 1919

Ornithodoros occidentalis Trape, Diatta, Durand & Renaud, 2013 etc, etc, etc.

Please do not use the notation "Mus musculus domesticus (M. m. domesticus)" - the abbreviation given in parentheses is not necessary and has no justification.

Thank you for your comment we have corrected the correct notation of the scientific names of the species and also remove the abbreviation given in brackets of Mus musculus domesticus (M. m. domesticus).